# Potential Decontamination of Drinking Water Pathogens through k-Carrageenan Integrated Green Bottle Fly Bio-Synthesized Silver Nanoparticles

**DOI:** 10.3390/molecules25081936

**Published:** 2020-04-22

**Authors:** M. A. Abu-Saied, Mohamed Elnouby, Tarek Taha, Muhammad El-shafeey, Ali G. Alshehri, Saad Alamri, Huda Alghamdi, Ali Shati, Sulaiman Alrumman, Mohamed Al-Kahtani, Mahmoud Moustafa

**Affiliations:** 1Polymer Materials Research Department, Advanced Technology and New Materials Research Institute, City of Scientific Research and Technological Applications (SRTA-City), New Borg El-Arab City, Alexandria 21934, Egypt; mouhamedabdelrehem@yahoo.com; 2Composite and Nanostructured Materials Research Department, Advanced Technology and New Materials Research Institute, City of Scientific Research and Technological Applications (SRTA-City), New Borg El-Arab City, Alexandria 21934, Egypt; m_nano2050@yahoo.com; 3Environmental Biotechnology Department, Genetic Engineering and Biotechnology Research Institute (GEBRI), City of Scientific Research and Technological Applications (SRTA-City), New Borg El-Arab City, Alexandria 21934, Egypt; t.h.taha@gmail.com; 4Department of Medical Biotechnology, Genetic Engineering and Biotechnology Research Institute (GEBRI), City for Scientific Research and Technology Applications (SRTA-City), New Borg El-Arab City, Alexandria 21934, Egypt; mohamedshafeey2010@yahoo.co.uk; 5Department of Biology, College of Science, King Khalid University, Abha 9004, Saudi Arabia; alialshri55@ymail.com (A.G.A.); saralomari@kku.edu.sa (S.A.); hudaghamdi@kku.edu.sa (H.A.); aaalshati@kku.edu.sa (A.S.); salrumman@kku.edu.sa (S.A.); dr.malkahtani@gmail.com (M.A.-K.); 6Prince Sultan Bin Abdulaziz Center for Environmental and Tourism Research and Studies, King Khalid University, Abha 9004, Saudi Arabia; 7Department of Botany and Microbiology, Faculty of Science, South Valley University, Qena 83523, Egypt

**Keywords:** green bottle fly, k-carrageenan, silver nanoparticles, antimicrobial activity

## Abstract

The wide distribution of infections-related pathogenic microbes is almost related to the contamination of food and/or drinking water. The current applied treatments face some limitations. In the current study, k-carrageenan polymer was used as supporting material for the proper/unreleased silver nanoparticles that showed strong antimicrobial activity against six pathogenic bacteria and yeast. The bio-extract of the pupa of green bottle fly was used as the main agent for the synthesis of silver nanoparticles. The qualitative investigation of biologically synthesized silver nanoparticles was determined using UV-Vis spectrophotometric analysis; however, the size of nanoparticles was in range of 30–100 nm, as confirmed by scanning electron microscopy (SEM), transmission electron microscopy (TEM), and particle size analyzer. The proper integration of silver nanoparticles into the polymeric substrate was also characterized through fourier transform infrared (FT-IR), thermogravimetric analysis (TGA), SEM, and tensile strength. The antimicrobial activity of k-carrageenan/silver nanoparticles against Gram positive, Gram negative, and yeast pathogens was highly effective. These results indicate the probable exploitation of the polymeric/nanoparticles composite as an extra stage in water purification systems in homes or even at water treatment plants.

## 1. Introduction

Many technological fields have incorporated metal nanoparticles in their applications, such as microelectronics [1], sensor applications [2,3], magnetic devices [4], dye removal [5], and antimicrobial activities [6]. Silver nanoparticles are one of the most interesting metal nanoparticles that have unique properties of being optically and catalytically active, showing elevated thermal and electrical conductivities. These properties are strongly dependent on the morphology of the formed silver nanoparticles [7,8]. Recent medical approaches used silver nanoparticles as antibacterial and antifungal agents as a result of their higher antimicrobial activities. They almost have a large surface area and small size (smaller than 20 nm), in addition to their high dispersion.

Various methods such as electrochemical [9] and green approaches have been used for the preparation of silver nanoparticles. Such green approaches use plant and microbial extracts as raw materials to act as reducing agents. The green synthesis of nanoparticles has more advantages than other reported methods. The produced nanoparticles are almost homogenous in their shapes and have narrow size distributions. In addition, the green methods are ecofriendly and cost-effective methods that do not produce toxic residues [10,11].

Carrageenans are major groups of galactans presented in red seaweeds [12]. They are natural polymers with hydrophilic properties [13], which are formed of linear chains of partially sulphated galactans that have a high tendency to form polymeric films (Figure 1). They have been extracted from the cell walls of various red seaweeds [14] such as *Kappaphycus alvarezii* and *Eucheuma denticulatum* [15]. In 1862, Stanford used the name Carraigin for the first time to describe the extract of *Chondrus crispus*. These sulphated polysaccharides have been used since 400 AD in Ireland as a curing agent for coughs and colds in addition to their usage as a gelation agent. The term “carrageenan” is more recent and is currently used by several authors since 1950 [16,17].

Carrageenans have many applications in food processes and even non-food industries and are considered as a high value functional ingredient in food [16,18]. They can be used as a stabilizer [19] in dairy products such as flavored milks [16]. They can also be used in many kinds of food including water-based foods, meat products [20], infant food, pet food [15], and nutritional supplement beverages. They are also used in cosmetics, pharmaceuticals, textile industries, and printing d. However, toxicological properties of carrageenans that have been used in many applications were shown at high doses [21,22,23].

The use of silver nanoparticles integrated with different polymeric materials for decontamination of pathogens in water has been previously reported. A mixture of Polyvinyl alcohol (PVA) and alginate was prepared and integrated with bacterial synthesized silver nanoparticles as antimicrobial composite for removal of pathogenic microbes from water [24]. Moreover, a composite of green synthesized silver nanoparticles and PVA/chitosan polymers has also been used for ceasing the microbial strains existing in water sample [25].

The current study concerns the proper exploitation of biosynthesized silver nanoparticles in the preparation of polymeric membrane that bears antimicrobial activity to be used as an extra stage in water purification systems for ceasing the growth of contaminating microbes. The antimicrobial activity of carrageenan integrating biosynthesized silver nanoparticles was tested and evaluated. In addition, the required characterizations of the prepared nanoparticles and the prepared polymeric membranes were also investigated in order to evaluate their successful preparation and their probable industrial applications.

## 2. Results and Discussion

### 2.1. Characterization of Silver Nanoparticles

#### 2.1.1. UV-Vis Spectrophotometry

UV-Vis spectroscopy is considered a simple, quick, and sensitive technique for investigating the formation of silver nanoparticles, which could be observed through the change in color from light yellow to dark brown [26,27]. The spectrophotometric analysis of the prepared silver nanoparticles was performed using a wide range of scanned wavelengths (200–900 nm). As shown in Figure 2, the highest absorbance was observed at over 400 nm, indicating the presence of silver nanoparticles. However, silver nitrate solution showed an absorbance peak at 300 nm that completely disappeared after the formation of silver nanoparticles. These data are almost matched with those of Ibrahim et al., who showed that the silver nanoparticles absorb the visible spectra at 409 nm accompanied the disappearance of the silver nitrate peak that was shown at 300 nm. Furthermore, previous studies reported that the peaks of UV absorption of silver nanoparticles could be in the range of 400–450 nm [28,29], which is in agreement with the data of the current study.

#### 2.1.2. Scanning Electron Microscopy (SEM)

SEM is an effective method to investigate the surface image of an object. It can precisely demonstrate the particle size, shape, and distribution of the tested material, in addition to the determination of its morphological appearance and investigating if it is in micro or nanoscale [30].

The scanning electron microscope of the current synthesized silver nanoparticles revealed that the formed particles are spherical shapes with a tendency to be aggregated. As shown in Figure 3, the formed rounded shapes are all in nanometer; however, the particle’s diameter was almost in the range of 22 nm to 82 nm. The variation in particles’ size indicates the development of silver nanoparticles [31].

#### 2.1.3. Transmission Electron Microscopy (TEM)

The biosynthesized silver nanoparticles were found to be spherical and polygon nanoparticles, which were monodispersed (Figure 4). In general, the shape and size of formed nanoparticles are almost dependent on the stabilizing agents of the medium reducing agents, which involves bottom-up synthesizing of silver nanoparticles [32]. The sizes of the obtained nanoparticles were in the range of 30–100 nm, which is in agreement with the results obtained by SEM micrographs (Figure 3), as also reported by Bhakya and his colleagues [33].

#### 2.1.4. Particle Size Analysis (PSA)

Particle size was detected based on measuring the time-dependent fluctuation of scattering of laser light by the nanoparticles in colloidal solution undergoing Brownian motion. Figure 5 shows that the main particle size was about 219 nm with some post aggregation up to 2800 nm. However, the particle size measured by other microscopic techniques, such as SEM and TEM, showed a much smaller particle size (Figure 3 and Figure 4). Prathna and his colleagues explained that the increased particle size could be owing to the bio-organics present in the preparing media, or could also be the result of other interaction forces in the solution like van der Waals forces of attraction [34].

### 2.2. Characterization of k-Carrageenans/Silver Nanoparticles Film

#### 2.2.1. Mechanical Properties

The formulated polymer (k-carrageenan’s film) produces transparent, flexible, and uniform film owing to its viscoelastic nature [35,36]. The max force, max displacement, max stress, and max strain measurements of k-carrageenan films prepared with and without silver nanoparticles are shown in Table 1.

The prepared film of k-carrageenan with silver nanoparticles exhibited higher strength with a higher percentage of elongation, stress, and strain compared with the film lacking silver nanoparticles. This can be explained in terms of the positive effect of silver nanoparticles on the polymeric network structure, as well as the polymeric network performance of the formed film.

The mechanical properties of carrageenan film without and with silver nanoparticles changed from 14.37 to 17.21 (N), 0.60 to 0.66 (mm), 17.96 to 18.15 (N/mm^2^), and 1.98 to 2.25 (%) for max force, max displacement, max stress, and max strain, respectively. This means that the addition of silver nanoparticles could improve the structural conformation of carrageenan gel, and thus improve the functional properties.

#### 2.2.2. Fourier Transform Infrared (FT-IR)

FT-IR spectra of the films consisting of k-carrageenan or k-carrageenan with silver nanoparticles are shown in Figure 6. The characteristic absorption bands of k-carrageenan at 843, 922, 1064, and 1251 cm^−1^ can be attributed to d-galactose-4-sulfate, 3,6-anhydro-D-galactose, glycosidic linkage, and ester sulfate stretching, respectively (Figure 6A). The broad band at 3200–3600 cm^−1^ is attributed to stretching absorption of -OH groups of k-carrageenan. However, the FT-IR spectrum of the nano-composite hydrogel film shows an absorption band at 596 cm^−1^, which can be related to the presence of silver nanoparticles (Figure 6B) [37].

#### 2.2.3. Thermogravimetric Analysis

Thermogravimetric analysis (TGA) is widely used to investigate the thermal decomposition of polymers. The TGA thermograms of the films composed of k-carrageenan and k-carrageenan with silver nanoparticles, at a heating rate of 20 °C/min under nitrogen atmosphere, are shown in Figure 7. The composite has a thermal stability higher than the k-carrageenan polysaccharide, which might be related to the reaction and crosslinking of the k-carrageenan and silver nanoparticles (Figure 7). According to the figure, the composite including silver nanoparticles exhibit more excellent thermal stability than its silver nanoparticles free, so that the mass loss of the prepared composite moved to a higher temperature. Here, after the degradation stages, the composite produces about 36% weight loss, whereas the loss for the free silver nanoparticles is higher (40%) at the same temperature [38]. In fact, the presence of silver nanoparticles in the k-carrageenan polymer composite causes more interaction of the functional groups of the prepared composite through further intermolecular crosslinking, as indicated by the results of the FT-IR study. It can be suggested that such interactions can lead to the formation of intermolecular cross-links between the k-carrageenan chains, and as a result, the mobility of the k-carrageenan chains could be reduced, thus improving the thermal stability of the resulting product. This result is consistent with the reported previous observations by Wei et al. [39,40]

The weight reductions at 600 °C are about 64% and 58% for k-carrageenan and k-carrageenan with silver nanoparticles, respectively. TGA thermograms of the samples show the main step of degradation, and the maximum rates of degradation are at 205 and 270 °C for k-carrageenan and k-carrageenan with silver nanoparticles, respectively. According to the SEM analysis of polymer/silver nanoparticles’ film, the nanoparticles are quite well distributed, which in turn enhances the overall thermal stability of the film rather than the plain polymeric film.

#### 2.2.4. Scanning Electron Microscopy (SEM)

SEM micrographs of the surface and cross-section of k-carrageenan films with and without silver nanoparticles were investigated. As shown in Figure 8A, a rough structure was detected in k-carrageenan before the addition of silver nanoparticles. However, these structures completely disappeared after the addition of silver nanoparticles, which indicates the homogenous formation of the polymer/silver nanoparticles composite (Figure 8B). Both films have an approximate thickness of 65 µm.

On the other hand, the smooth surface with few insoluble particles of plain k-carrageenan was investigated (Figure 8C) compared with the rough surface after the addition of silver nanoparticles, indicating the good distribution of silver nanoparticles on the surface of the hydrogel, as clearly observed (Figure 8D).

#### 2.2.5. Inductively Coupled Plasma (ICP)

The polymer/silver nanoparticles films are firstly submitted to surface washing, which resulted in the removal of unbounded silver nanoparticles. The ICP results of the washed membrane revealed the absence of silver nanoparticles from the external aqueous solution, which confirms that the internal nanoparticles are not released to the external contacting solution.

### 2.3. Antimicrobial Activity

The antimicrobial activity of k-carrageenan, k-carrageenan/pupa extract, and k-carrageenan integrated silver nanoparticles against six microbial pathogens was investigated using the disc diffusion method. As shown in Table 2, no antimicrobial activity was detected for k-carrageenan or k-carrageenan/pupa extract film; however, k-carrageenan integrated silver nanoparticles’ film showed a noticeable potent antimicrobial behavior against all tested bacterial and yeast strains. These results indicate that the observed antimicrobial activity is exclusive for silver nanoparticles and plain k-carrageenan or pupa extract are almost inactive, which is matched with other published data in the literature [41].

The potency of k-carrageenan integrated silver nanoparticles’ discs was approximately microbe-dependent. The most affected pathogens were *Pseudomonas aeruginosa* ATCC9027 and *Klebsiella pneumoniae* ATCC13883, which showed 16 mm clear zones, followed by *Vibrio cholerae* ATCC700 strain, which showed 15 mm, compared with *Escherichia coli* NCTC10418 strain, which was the least affected microbe (12 mm clear zone). However, *Candida albicans* ATCC 700 and *Bacillus cereus* ATCC6633 strains were moderately affected, with modest clear zone measurements (14 mm).

The ability of silver nanoparticles to cease the microbial growth is probably related to the interaction of silver nanoparticles with the thiol groups in the microbial cell walls, as reported by Zhang and his colleagues. They also reported that the fungal and bacterial cells were structurally changed after their contact with silver nanoparticles [42].

Additionally, the high surface area of silver nanoparticles allows them to contact better with microbial cells compared with bulk silver. Moreover, two successive steps were proposed for the mechanism of action of silver nanoparticles against pathogenic microbes. The first step includes the accumulation of silver nanoparticles in the microbial cell walls followed by the formation of pits in the walls, which leads to microbial death. The second step includes the penetration of silver nanoparticles into the bacterial of fungal cells with binding to the cell walls and membranes and, eventually, inhibition of the respiration process [42].

In the end, the antimicrobial activity of k-carrageenan integrated silver nanoparticles could boost their application in the decontamination of pathogenic microbes from drinking water. This polymeric/nanoparticles composite could be prepared in the form of cylindrical membranes to be added as an extra stage in water purification systems in homes or even at water treatment plants.

#### 2.3.1. Antimicrobial Activity of k-Carrageenan/Silver Nanoparticles Film against Real Water Sample

The ability of the prepared polymer/nanoparticles film to stop the probable pathogenic microbes in real drinking water sample was investigated. As shown in Figure 9, the discs containing tube showed an optical density close to zero compared with the discs-free tubes, which showed heavy growth of the inhabiting microbes. These results indicate that the film of k-carrageenan/silver nanoparticles has potent antimicrobial activity against microbes invading drinking water, and supports its application as an extra stage in water purification systems in homes or even at water treatment plants.

#### 2.3.2. Qualitative Determination of Some Bacterial Pathogens in Water

The ability of k-carrageenan/silver nanoparticles discs to affect the growth of water samples collected from the industrial zone of Borg Elarab city was determined. As shown in Table 3, the total plate count of mix 1 sample was recorded as 6.39 × 10^5^ cfu/mL, which was reduced to 5.3 × 10^4^ cfu/mL after treatment with the polymer/silver nanoparticles discs. Meanwhile, the Coliform and *E. coli* were completely absent from the same sample before the treatment process. On the other hand, the second sample was more contaminated with microbial strains. The k-carrageenan/silver nanoparticles discs succeeded to minimize the bacterial count of sample mix 2 from 9.17 × 10^5^, 44 × 10^5^, and 172 × 10^5^ to 5.2 × 10^4^, 1600, and 90 cfu/mL for total plate count (TPC), coliform, and *E. coli*, respectively. These results indicate that the prepared k-carrageenan/silver nanoparticles film can be successfully used for the treatment of bacterial pathogens in water samples. It worth mentioning that the bacterial count in all samples was investigated using colony forming unit (cfu/mL).

## 3. Materials and Methods

### 3.1. Materials

K-carrageenan was obtained from Sigma-Aldrich Chemicals, Ltd. (Schnelldorf, Germany), and green bottle fly (*Lucilia sericata*) was kindly provided by Entomology Department, Faculty of Science, Alexandria University (Alexandria, Egypt).

### 3.2. Preparation of Insects’ Pupa

The whole insects were reared in a specific cage using fresh rabbit’s liver as a feeding source and an egg hatching supporter. After the eggs’ incubation period, both the larvae and pupa were successively formed, and the latter one was used for the following experiments.

### 3.3. Extraction of the Metabolic Contents of Pupa

The formed pupas were collected from the insects’ cages, transferred into 50 mL sterile falcon tubes, and kept at −20 °C for three days. After freezing, the contents of pupa were extracted through the grinding in a sterile pestle till a fine powder was obtained. The powder was weighted to 10 g and was added to 100 mL sterile phosphate buffer saline (PBS) (pH 7.0). The PBS-suspended particles were incubated at room temperature and shaken at 150 rpm for 24 h. The incubation process was responsible for the dissolution of the water-soluble contents into the used buffer. After incubation, the suspended solution was filtered through 0.4 µm filter system, where the filtrate was rich with metabolic active ingredients of pupa. A specific weight of AgNO_3_ was added to 50 mL of the filtrate to obtain a final concentration of 1 mM. The solution was subsequently incubated at 30 °C and 200 rpm till the formation of brown color, which indicates the formation of silver nanoparticles.

### 3.4. Separation of the Obtained Silver Nanoparticles

The green synthesized silver nanoparticles were transferred into a 50 mL falcon tube and centrifuged at 15,000 rpm for 30 min. The collected brown pellet was washed twice with absolute ethanol and once with distilled water. The pellet was then dried at 50 °C for 24 h and kept dry till the following experiments.

### 3.5. Spectrophotometric Characterization of Silver Nanoparticles

Both silver nitrate (10 mM) and the obtained silver nanoparticles suspended in distilled water were scanned through the UV-Visible absorbance from 200 to 900 nm to investigate the highest peak of absorbance at a wavelength specified for the presence of silver ions and silver nanoparticles.

### 3.6. Preparation of k-Carrageenan Film Amended with Silver Nanoparticles

K-carrageenan solution (4% *w*/*v*) was prepared by dissolving 4 g of k-carrageenan in 100 mL distilled water (pH 7.0) at 80 °C with stirring on a magnetic stirrer (400 rpm). Silver nanoparticles were added to the solution at a concentration of 25 mg/50 mL. All mixtures were blended by stirring on a magnetic stirrer (400 rpm) for 3 h. According to the thermo-reversible characteristics of carrageenan gel, the formation mechanism of the gel is mostly dependent on temperature and other gel-inducing agents.

The existence of carrageenan at a high temperature (above 80 °C) is responsible for its random coil structure, which resulted from the electrostatic repulsions between neighboring chains. However, the reduction of the temperature will result in changing of the chains’ conformation into a helical structure. Moreover, the cooling conditions and the presence of cations can help the intermolecular interactions among the carrageenan chains and resulted in the agglomeration of the helical dimers forming a stable three-dimensional network [43]. Finally, both plain polymer and polymer integrated silver nanoparticles samples were poured onto polystyrene petri dishes and kept at room temperate till the formation and drying of films. Both films were washed twice using distilled water after drying.

### 3.7. Characterization of k-Carrageenan Films with/without Silver Nanoparticles

Fourier transform infrared (FT-IR) spectroscopy (Shimadzu FTIR-8400 S, Tokyo, Japan) was used to identify the presence of sulfonic acid groups in selected k-carrageenan and to confirm the chemical structure of k-carrageenan/silver nanoparticles. Thermo gravimetric analysis (TGA, Shimadzu TGA-50, Tokyo, Japan) was used to determine the weight loss and thermal degradation of the k-carrageenan before and after the addition of silver nanoparticles. Surface and internal cross-section imaging of plain k-carrageenan, silver nanoparticles, and k-carrageenan/silver nanoparticles’ films were performed using scanning electron microscopy (JEOL, JSM-6360 LA, Tokyo, Japan). The dried samples were coated with a thin gold layer using the sputtering technique. More characteristic analyses were performed for silver nanoparticles using transmission electron microscopy (TEM, JEOL (JEM-2100 plus, Tokyo, Japan), and the obtained samples were dispersed in ethanol and sonication for 15 min at room temperature. The particle size analyzer (PSA, Beckman Coulter, California, CA, USA) was also performed for silver nanoparticles. Tensile strength was carried out for k-carrageenan and k-carrageenan/silver nanoparticles’ films using universal testing machine (Shimadzu UTM, Tokyo, Japan). Moreover, the concentration of silver nanoparticles that might be released from the polymeric film after contacting with external aqueous solution was investigated using inductively coupled plasma (ICP, model: ICP-OES Prodigy, Teledyne Leeman Labs, Ohio, OH, USA).

### 3.8. Antimicrobial Activity of k-Carrageenan Films with/without Silver Nanoparticles

The antimicrobial activity of k-carrageenan film before and after the addition of silver nanoparticles in addition to k-carrageenan loaded pupa extract was investigated against six human microbial pathogens. The plain k-carrageenan film was 4% concentration, while the film of k-carrageenan integrated silver nanoparticles included 50 mg/100 mL of the prepared nanoparticles. At first, Luria Bertani (LB) agar was prepared according to the manual instructions, and was then sterilized by autoclaving at 15 psi and 121 °C for 15 min. After sterilization, the agar medium was poured into sterile petri plates and kept at 4 °C till solidification. Overnight LB broth cultures of the strains *Pseudomonas aeruginosa* ATCC9027, *Escherichia coli* NCTC10418, *Bacillus cereus* ATCC6633, *Candida albicans* ATCC 700, *Vibrio cholerae* ATCC700, and *Klebsiella pneumoniae* ATCC13883 were prepared under aseptic conditions. Each microbial pathogen was diluted using sterile 0.9% NaCl solution to obtain 0.5 McFarland standards. A sterile cotton swab of each diluted pathogen was aseptically spread over the LB plates, followed by the surface addition of k-carrageenan, k-carrageenan/pupa extract, and k-carrageenan/silver nanoparticles discs. The plates were incubated at 30 °C for 24 h. The formed clear zones were measured and recorded.

### 3.9. Antimicrobial Activity of k-Carrageenan/Silver Nanoparticles against Drinking Water Sample

The antimicrobial activity of k-carrageenan/silver nanoparticles film was also tested against the probable microbes of real drinking water sample. Two sterile falcon tubes were filled with 5 mL of sterile LB broth under aseptic conditions. Both tubes were inoculated with 100 µl of drinking water sample. One of the tubes was amended with three discs (0.7 mm) of k-carrageenan/silver nanoparticles film and incubated at 30 °C and 150 rpm for 24 h. After incubation, the optical densities of both tubes were measured at 600 nm against un-inoculated LB broth.

### 3.10. Qualitative Determination of Some Bacterial Pathogens in Water

The ability of the prepared k-carrageenan/silver nanoparticles film to cease the growth of bacterial pathogens in water samples was investigated. Two water samples named Mix1 and Mix2 were collected from the industrial zone of Borg Elarab city. Both samples were distributed into sterile falcon tubes (10 mL each), in which two portions of each sample acted as the control and treatment samples. Each treatment sample was exposed to one disc of the prepared film (7 mm diameter) under sterile conditions. All the prepared control and treatment samples were incubated at 30 °C and 150 rpm for 24 h. After incubation, the total plate count (TPC), *E. coli*, and Coliform count were qualitatively determined. The determination of *E. coli* and Coliform was achieved through the inoculation of 1/100 diluted samples into ready-to-use plates of Compact Dry “Nissui” EC for Coliform and *E. coli* (NISSUI Pharmaceutical Co., LTD., Tokyo, Japan).

## 4. Conclusions

Silver nanoparticles were synthesized using the green method. The shape and size of the prepared nanoparticles were confirmed using various characteristic techniques including UV-Vis spectrophotometry, SEM, TEM, and PSA. Different characterization techniques were used including SEM, FT-IR, TGA, and tensile properties for both plain polymer and silver nanoparticles-integrated polymer. k-carrageenan/silver nanoparticles’ film was tested for its antimicrobial activity, and showed a detectable antimicrobial activity against six human pathogenic bacteria and yeast, in addition to real water samples, which boosts its probable application as an extra stage in water purification systems.

## Figures and Tables

**Figure 1 molecules-25-01936-f001:**
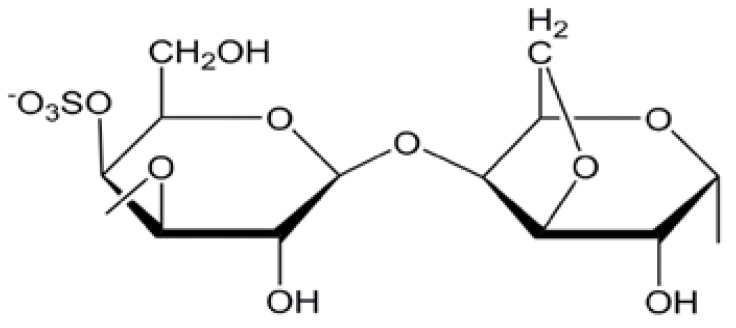
Chemical structure of k-carrageenan.

**Figure 2 molecules-25-01936-f002:**
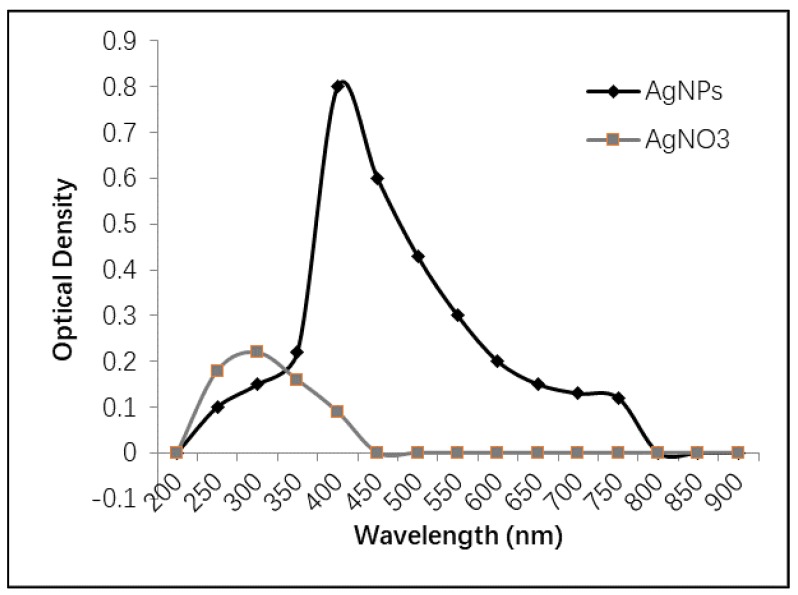
UV-Vis absorption spectra of silver nanoparticles (AgNPs) and AgNO_3_.

**Figure 3 molecules-25-01936-f003:**
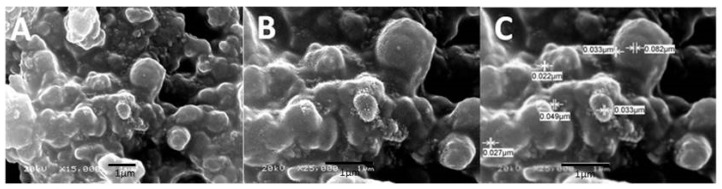
Scanning electron microscopy (SEM) of biosynthesized silver nanoparticles: (**A**) micrograph of silver nanoparticles at magnification 15,000×, (**B**) micrograph of silver nanoparticles at magnification 25,000×, and (**C**) measured diameters of the formed silver nanoparticles in micrometer.

**Figure 4 molecules-25-01936-f004:**
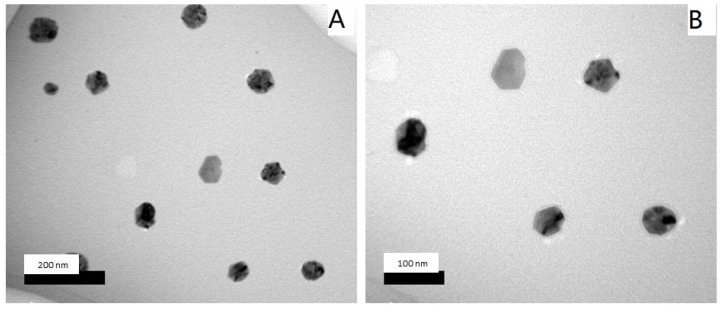
Transmission electron microscopy (TEM) micrographs of silver nanoparticles at (**A**) low (100,000×) and (**B**) high magnifications (200,000×).

**Figure 5 molecules-25-01936-f005:**
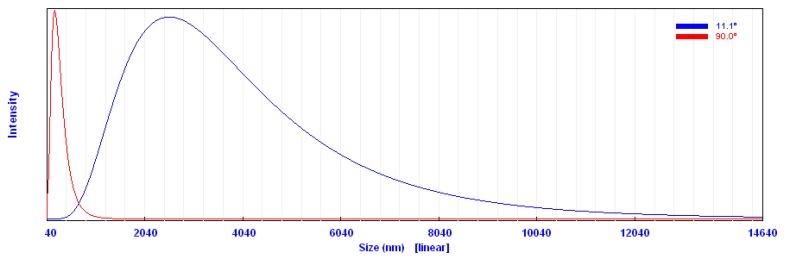
Particle size histogram of prepared silver nanoparticles.

**Figure 6 molecules-25-01936-f006:**
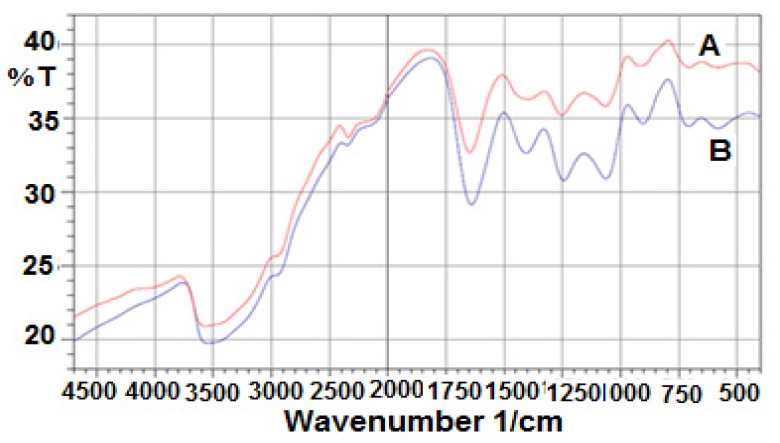
Fourier transform infrared (FT-IR) spectra of (**A**) k-carrageenan and (**B**) k-carrageenan with silver nanoparticles.

**Figure 7 molecules-25-01936-f007:**
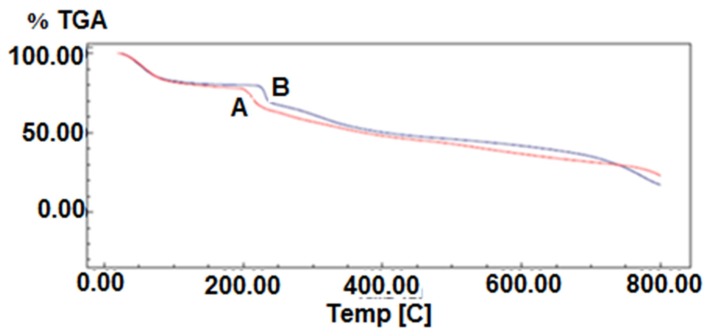
Thermogravimetric analysis (TGA) curves of (**A**) k-carrageenan and (**B**) k-carrageenan with silver nanoparticles.

**Figure 8 molecules-25-01936-f008:**
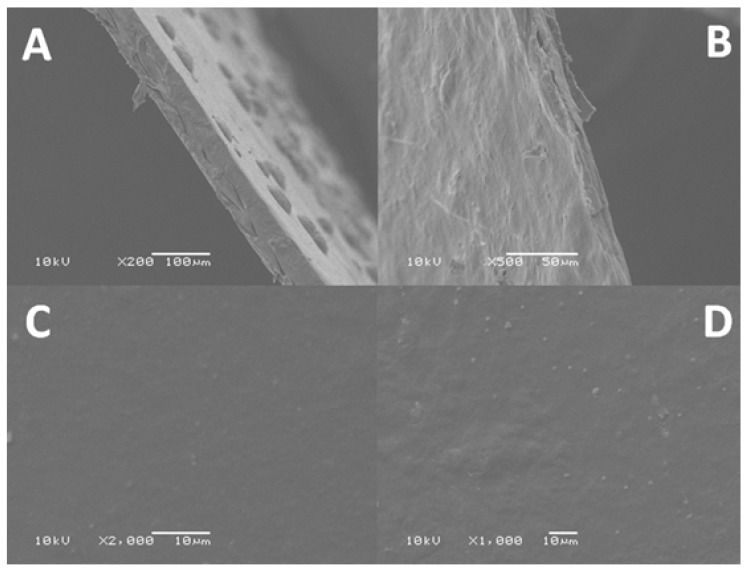
SEM of surface and cross-section of k-carrageenans with and without silver nanoparticles: (**A**) cross section of k-carrageenans, (**B**) cross section of k-carrageenans with silver nanoparticles, (**C**) surface of k-carrageenans, and (**D**) surface of k-carrageenans with silver nanoparticles.

**Figure 9 molecules-25-01936-f009:**
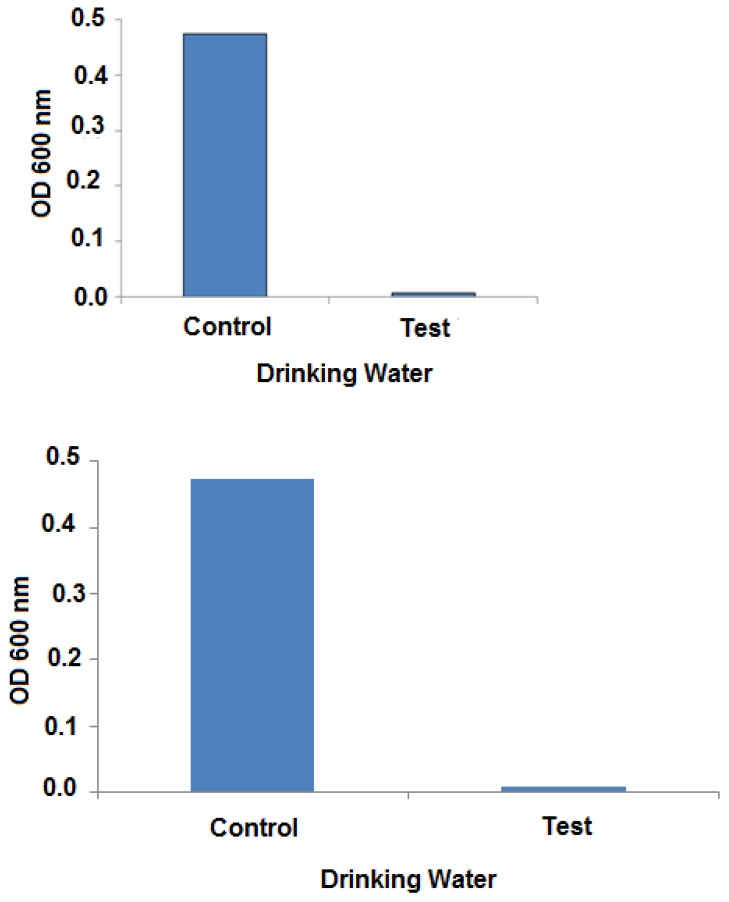
Antimicrobial activity of k-carrageenan/silver nanoparticles discs against microbes invading drinking water. Control: discs-free water sample and test: discs-amended water sample cultivated in Luria Bertani (LB) broth. OD: optical density.

**Table 1 molecules-25-01936-t001:** Mechanical properties of k-carrageenan with and without silver nanoparticles.

Sample	Max Force (N)	Max Displacement (mm)	Max Stress (N/mm^2^)	Max Strain (%)
k-carrageenan	14.37 ± 0.15	0.60 ± 0.02	17.96 ± 0.17	1.98 ± 0.01
k-carrageenan with silver nanoparticles	17.21 ± 0.15	0.66 ± 0.02	18.15 ± 0.17	2.25 ± 0.01

**Table 2 molecules-25-01936-t002:** Antimicrobial activities of k-carrageenan and k-carrageenan integrated silver nanoparticles against human pathogenic microbes.

Pathogenic Microbes	Clear Zones (mm)		
	k-Carrageenan	k-Carrageenan/Pupa Extract	k-Carrageenan/Silver Nanoparticles
*Vibrio cholerae*	0	0	15
*Candida albicans*	0	0	14
*Pseudomonas aeruginosa*	0	0	16
*Escherichia coli*	0	0	12
*Klebsiella pneumoniae*	0	0	16
*Bacillus cereus*	0	0	14

**Table 3 molecules-25-01936-t003:** Testing of k-carrageenan/silver nanoparticles discs on total plate count (TPC), Coliform, and *E. coli* of two water samples collected from the industrial zone of Borg Elarab city.

Sample	TPC (CFU/mL)	Coliform (CFU/mL)	*E. coli* (CFU/mL)
Mix 1 (Control)	6.39 × 10^5^	Absent	Absent
Mix 1 (Treated)	5.3 × 10^4^	Absent	Absent
Mix 2 (Control)	9.17 × 10^5^	44 × 10^5^	172 × 10^5^
Mix 2 (Treated)	5.2 × 10^4^	1600	90

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
