# Peer review of "Potential Decontamination of Drinking Water Pathogens through k-Carrageenan Integrated Green Bottle Fly Bio-Synthesized Silver Nanoparticles"

_molecules, 2020, doi:10.3390/molecules25081936_

Round 1
Reviewer 1 Report
General comments:
After corrections some of the problems still need to be improved:
Please refer to the possible high swelling of obtained films in the contact with aqueous external solutions. Is it possible that in this environment silver nanoparticles will be released to the external media?
Based on the FTIR spectra provided an absorption band in 400 cm−1 can not be detected, and even if, it is hard to see the difference between both (non-modified, and Ag-modified) films.
“The composite has thermal stability higher than the κ-carrageenan polysaccharide which might be related to the reaction and crosslinking of the κ-carrageenan and silver nanoparticles.” – explain how it is possible. Moreover, explain why the thermal behavior within 20-800 °C need to be tested? How it influences the possible application of this material? A discussion of TGA results should be improved.
Particular comments:
The X-axis in Figure 7 should end at 100%.
Author Response
Dear reviewers,
At first, we would like to thank you for the efforts expended to revise the corrections of the first round of the current manuscript, and we are happy to respond to your mentioned comments as follows:
Reviewer 1:
-Please refer to the possible high swelling of obtained films in the contact with aqueous external solutions. Is it possible that in this environment silver nanoparticles will be released to the external media?
Response:
The authors confirm that the polymer/silver nanoparticles films are firstly submitted to surface washing that resulted in the removal of unbounded silver nanoparticles. The inductively coupled plasma (ICP) results of the washed membrane revealed the absence of silver nanoparticles from the external aqueous solution, which confirms that the internal nanoparticles are not released to the external contacting solution.
These observations have been added to the manuscript in section 2.2.5. and line 197 (Results section) and line 319 (methods section).
-Based on the FTIR spectra provided an absorption band in 400 cm−1 cannot be detected, and even if, it is hard to see the difference between both (non-modified, and Ag-modified) films.
Response:
The FTIR band of silver nanoparticles has been revised and corrected in the manuscript (Line 162-164).
-“The composite has thermal stability higher than the κ-carrageenan polysaccharide which might be related to the reaction and crosslinking of the κ-carrageenan and silver nanoparticles.” – explain how it is possible.
Response:
The composite has a thermal stability higher than the k-carrageenan polysaccharide which might be related to the reaction and crosslinking of the k-carrageenan and silver nanoparticles in Figure 7. According to the figure, the composite including silver nanoparticles exhibits more excellent thermal stability than its free silver nanoparticles, so that the mass loss of the prepared composite moved to a higher temperature. Here, after degradation stages, the composite produces about 36% weight loss, whereas the loss for the free silver nanoparticles is higher (40%) at the same temperature [38]. In fact, the presence of silver nanoparticles in the k-carrageenan polymer composite causes more interaction of the functional groups of the prepared composite through further intermolecular crosslinking as indicated by the results of the FTIR study. It can be suggested that such interactions can lead to the formation of intermolecular cross-links between the k-carrageenan chains, and as a result, it could be the reduced mobility of the k-carrageenan chains and thus improving the thermal stability of the resulting product. This result is consistent with the reported previous observations by Wei et al. [39,40].
These observations have been added to the manuscript in section 2.2.3. Thermogravimetric analysis and line 171 to 183.
-Moreover, explain why the thermal behavior within 20-800 °C need to be tested? How it influences the possible application of this material? A discussion of TGA results should be improved.
Response:
Some authors of this manuscript are working in Polymer Materials Research Department and Composite and Nanostructured materials research department. So, the main interests of those authors are to study the properties and behavior of materials of any research applications. On the other hand; thermo gravimetric analysis (TGA, Shimadzu TGA-50) was used to determine the weight loss and thermal degradation of the k-carrageenan before and after the addition of silver nanoparticles with behavior within 0-800 °C because the condition of the sample was determined before the test.
-The X-axis in Figure 7 should end at 100%.
Response:
The y- axis in figure 7 has been changed to be ended at 100%.

Reviewer 2 Report
Manuscript deals with the use of silver nanoparticles synthesized by green methods, even when authors considered most of the comments made by reviewers in the first round, and most of the queries were taken into account, there are still minor revisions to be considered:
Line 75. Revise writing, “However, toxicological properties of carrageenan have been shown at very high doses that do not have not used in many applications as previously demonstrated almost used in current applications”.
Figure 4. Please indicate what magnifications were used for Figures A and B.
Lines 154,157, 294, 296, 360. Please homogenize the term k-carrageenan along the text, ensure that the “k” is always used as small case letter.
Table 1. How many samples or repetitions were used for each treatment? Please provide the standard deviation and the statistical analysis for mean comparison.
Line 167. Revise if “absorption at 400 cm-1” is a correct statement, since 400 is the edge of the Figure and where trend differences between treatments at this specific wavenumber are not so evident.
Figure 6. Indicate the axis label explicitly for the wavenumber instead of 1/cm.
Line 178. What is DTGA?
Lines 181-182. “According to the SEM analysis of polymer/silver nanoparticles' film, the nanoparticles are homogeneously distributed which in turn enhance the overall thermal stability of the film rather than the plain polymeric film”. But neither TGA nor SEM images provide sufficient information about homogeneous distribution of silver nanoparticles within the polymer matrix, maybe Raman spectroscopy could be more helpful for supporting this statement.
Line 194. Revise writing “traces un-soluble spheres”. This sentence is not clear.
Section 2.2. In the previous review it was required to change SEM images in order to present two figures at the same magnification, in order to perform a better comparison and discussion.
Line 153. Units format for “ufc” is different to that used un Table 3.
Section 3.3 It is still not clear which is the main reason for using “pupa extracts” for nanoparticles synthesis, or how these extracts act during nanoparticle formation or how they could act for improving or not, the functional activity as antimicrobial agent.
Table 2. A control with k-carrageenan and pupa extract without nanoparticles should be used, in order to explain if the improved performance of the composites is due to the size of silver nanoparticles, or pupa extract has any contribution or justification.
Author Response
Dear reviewers,
At first, we would like to thank you for the efforts expended to revise the corrections of the first round of the current manuscript, and we are happy to respond to your mentioned comments as follows:
Reviewer 2:
- Line 75. Revise writing, “However, toxicological properties of carrageenan have been shown at very high doses that do not have not used in many applications as previously demonstrated almost used in current applications”.
Response:
The sentence has been revised and rewritten.
- Figure 4. Please indicate what magnifications were used for Figures A and B.
Response:
The magnification of Figure 4A was 100,000X and the magnification of Figure 4B was 200,000X
Both magnifications have been added to the figure legend.
- Lines 154,157, 294, 296, 360. Please homogenize the term k-carrageenan along the text, ensure that the “k” is always used as small case letter.
Response:
The "k" in the term "k-carrageenan" has been revised along the whole document and rewritten in small case letter.
-Table1. How many samples or repetitions were used for each treatment? Please provide the standard deviation and the statistical analysis for mean comparison.
Response:
The numbers of samples are three for each treatment and the standard deviation and the statistical analysis for mean comparison have been added in table 1.
-Line 167. Revise if “absorption at 400 cm-1” is a correct statement, since 400 is the edge of the Figure and where trend differences between treatments at this specific wavenumber are not so evident.
Response:
The FTIR band of silver nanoparticles has been revised and corrected in the manuscript (Line 162-164).
-Figure 6. Indicate the axis label explicitly for the wavenumber instead of 1/cm.
Response:
Figure 6 was corrected by adding the phrase "wavenumber" to indicate the axis label explicitly instead of 1/cm.
-Line 178. What is DTGA?
Response:
Line185. DTGA has been corrected to TGA
- Lines 181-182. “According to the SEM analysis of polymer/silver nanoparticles' film, the nanoparticles are homogeneously distributed which in turn enhance the overall thermal stability of the film rather than the plain polymeric film”. But neither TGA nor SEM images provide sufficient information about homogeneous distribution of silver nanoparticles within the polymer matrix, maybe Raman spectroscopy could be more helpful for supporting this statement.
Response:
Thanks for the reviewer for the fruitful comment. The sentence has been rewritten in another way that indicates the well distribution of the silver nanoparticles inside the polymeric membrane as indicated by the SEM micrographs.
- Line 194. Revise writing “traces un-soluble spheres”. This sentence is not clear.
Response:
The words have been revised and rewritten more clearly.
-Section 2.2. In the previous review it was required to change SEM images in order to present two figures at the same magnification, in order to perform a better comparison and discussion.
Response:
The authors agree with the reviewer’s guidance but, unfortunately, additional measurements or characterization are not currently available since all universities and research institutes are closed as a result of the spread of the COVID-19 virus.
- Line 153. Units format for “ufc” is different to that used un Table 3.
Response:
We have checked the whole document and the unit format "ufc" is not exist.
-Section 3.3 It is still not clear which is the main reason for using “pupa extracts” for nanoparticles synthesis, or how these extracts act during nanoparticle formation or how they could act for improving or not, the functional activity as antimicrobial agent.
Response:
Most methods used for the synthesis of silver nanoparticles are related to plant extracts or microbial extracts. However, recent research is concerning by figuring out other effective methods for the synthesis of silver nanoparticles. Some of these methods are depending on using insects and their products, for instance: Propolis (bee glue) has been used for the synthesis of silver nanoparticles. Please see the following reference:
Hamed A. Ghramh, Khalid Ali Khan, Essam H. Ibrahim, and Mohd Javid Ansar (2019). Biogenic Synthesis of Silver Nanoparticles Using Propolis Extract, Their Characterization, and Biological Activities. Science of Advanced Materials Vol. 11, pp. 876–883.
From this point of view, the authors of the current manuscript proposed to screen the ability of blow fly pupa extract to synthesis silver nanoparticles. The authors assumed that this extract might have reducing agents that able to reduce silver ions into silver nanoparticles. Moreover, the author's expectation was true, since silver nanoparticles were already formed and their formation was confirmed using different characteristic techniques.
-Table 2. A control with k-carrageenan and pupa extract without nanoparticles should be used, in order to explain if the improved performance of the composites is due to the size of silver nanoparticles, or pupa extract has any contribution or justification.
Response:
A control with k-carrageenan and pupa extract without nanoparticles had been used and the results were added to table 2.

Round 2
Reviewer 1 Report
The manuscript has been updated according to my suggestions, thus it can be accepted in the present form.
This manuscript is a resubmission of an earlier submission. The following is a list of the peer review reports and author responses from that submission.
Round 1
Reviewer 1 Report
This manuscript deals with the use of silver nanoparticles synthesized by means of Lucilia sericata extract and their potential as antimicrobial agent in drinking water. Despite the subject in the title seems to be interesting and striking, there are some specific comments about M&M section that should be cleared for a better performance and reliability of R&D section. Some methodologies are lacking for specific procedures that may be relevant for the new knowledge creation, and some others should be considered for reinforcing the hypothesis stated. Manuscript lacks discussion, it is merely descriptive. According to the title of the manuscript, the reader would expect to find a successful application of a biopolymeric material loaded with Au nanoparticles as a drinking water decontaminant, but results shows that microbiological characterization lacks robustness, therefore it is not possible to establish if this biomaterial will achieve the main objective of this work, as stated in the title. I recommend considering great major revisions before being published.
Some specific comments are listed below
Line 66. Revise spacing in this line.
Section 2.1.1. Authors establish that a maximum absorbance at ⁓400 nm wavelength indicates the presence of silver nanoparticle formation, as previously reported by Aziz et al. [26]. Nevertheless, Aziz et al. (2017), indicate that this behavior is due to silver reduction from AuNO3 in presence of cationic biopolymer like chitosan. How do authors ensure that this maximum correspond to the same type of reaction? Why the OD axis is labeled at 600 nm in Figure 2?
Sections 2.1.2 & 2.1.3. Images in SEM and TEM micrographics should be improved, they are blurred. The scale bar is not clear. Authors state that silver nanoparticles tend to for aggregates in agreement to SEM images, but this behavior is not seen in TEM images, thus, are these differences due to the sample preparation procedure required in each technique or it is true that nanoparticles interact among them forming clusters? Please explain.
Section 2.1.4. Hydrodynamic diameter is a technique plenty used for nanoscale particles, like the synthesized silver nanoparticles of this manuscript, but it is not clear in which methods single nanoparticles are used and when carrageenan-loaded with nanoparticles is used. In this sense, could largest particle hydrodynamic particle be attributed to any carrageenan effect? What pH condition was used? Are silver nanoparticles soluble under this pH? Why do authors not used any control sample? Why do authors report two laser dispersion angles?
Section 2.2.1. Authors evaluated the mechanical properties of carrageenan-loaded with silver nanoparticles compared to pure carrageenan, but is there statistical differences between both treatments? This means, the addition of silver nanoparticles could improve the structural conformation of carrageenan gel, and therefore improve the functional properties?
Section 2.2.2 Authors stated that changes at 400 cm-1 wavelength number is attributed to the presence of silver nanoparticles, but the spectra displays 400 cm-1 as the merely limit of detection of the test, maybe other type of spectroscopy like XRD, could be more supportive to the FTIR, as many of the works related to metal nanoparticle synthesis.
Section 2.2.3. It is not clear at all, the real contribution of this section to the main objective of the manuscript, since authors stated that TGA analysis was carried out for investigating the thermal decomposition of polymers, but TGA of k-carrageenan is already reported, besides that any thermal treatment was applied to the silver nanoparticles loaded within carrageenan, therefore it is nonsense to inform these results. If authors stated that around 200 °C is carried out the thermal decomposition of carrageenan with/without silver nano particles, why do authors evaluate the thermal behavior until 800°C, and what do authors mean by carrageenan degradation? Since more than 30% of initial weight is recorded at the end of the test, and Silver nanoparticles does not account for this percentage?
Section 2.2.4. It is suggested to provide SEM images (C and D) at the same magnification than those in Section2.1.2, in order to have a better comparison.
Section not numbered but specified as “Antimicrobial activity”. It is suggested to use a more quantitative assay in order to support the qualitative one reported as the inhibition halo formation. Also, some SEM images to cell cultures exposed to silver nanoparticles alone and with carrageenan could provide more information about the microbial inhibition mechanisms that support the hypothesis stated in lines 188-191.
Lines 192-198. Raman spectroscopy or confocal microscopy could be helpful for visualizing the chemical and spatial distribution of silver nanoparticles in the microbial culture. Otherwise, it is not possible to ensure that this hypothesis is occurring.
Lines 199-212. Results in these lines are merely predictive, and do not have the sufficient support for ensuring that biopolymer disks loaded with silver nanoparticles inactivate the bacteria culture, neither measuring only the optical density as a unique methodology.
Lines 228-233. It is stated: “After incubation, the suspended solution was filtered through 0.4 μm filter system; where, the filtrate was rich with metabolic active ingredients of pupa. A specific weight of AgNO3 was added to 50 ml of the filtrate to obtain a final concentration of 1 mM. The solution was subsequently incubated at 30°C and 200 rpm till the formation of brown color that indicates the formation of silver nanoparticles.” What does authors mean by “metabolic active ingredients of pupa”? Could authors provide more bibliographic resources that plenty justified and described the mechanisms associated with pupa state of Lucilia sericata insect with the nanoparticle formation? How do these compounds act for forming the silver nanoparticles, since some references used by authors, state that they use some plant tissues for reducing the particle size of metal materials?
Section 3.4. What do authors mean with “green synthesized silver nanoparticles if previous paragraph state that the brownish color indicates their formation? See comment above, where previous reports in the literature indicate that brown color during silver nanoparticle formation is attributed to metal reduction.
Section 3.5. Please provide the UV-Vis spectra for AgNO3, is there any difference with silver nanoparticles?
Section 3.7. Please provide a brief description about the pre-treatments made to nanoparticles for SEM, TEM, hydrodynamic diameter characterizations.
Lines 284-289. How do authors ensure that this is the most convenient form to evaluate the microbial decontamination in drinking water? Maybe the UFC/mL quantification may support these results.
Conclusions. This section started with the use of “the pupa of green bottle fly as a bio-material for the bio-synthesis of silver nanoparticles”, but discussion about this statement is not covered in the document. Also it is stated that “These nanoparticles were successfully integrated into k-carrageenan, and their structure was characterized using SEM, FT-IR, TGA and tensile properties”, but there is not any quantitative technique that indicate the successfully integration of silver nanoparticles within the carrageenan matrix. Finally, it is stated “K-carrageenan/silver nanoparticles' film was tested for its antimicrobial activity, and showed a powerful activity against six human pathogenic bacteria and yeast, which boosts its probable application as an extra stage in water purification systems.” But any quantitative assay for living-cells was used, nor any control sample (negative and positive controls) was used, thus it is not possible to establish that this material displayed powerful activity against bacteria growth or survival.
Reviewer 2 Report
General comments:
In my opinion, these results are really interesting however there is a lack of several important things, like a) as silver-polymer systems were already tested a hundred of times in literature, why Authors choose this polymer, b) the way of acting of Ag-nanoparticles against different microbes is already know, thus it was also expected in this system. What constitutes the novelty of this research? Is there any problem to solve?
In my opinion, the title of this manuscript does not refer to the content as most of the presented data refers rather to the structure of Ag-nanoparticles and polymeric films. Only a small part of the data is devoted to the potential of polymeric films in water decontamination.
Please refer to the possible high swelling of obtained films in the contact with aqueous external solutions. Is it possible that in this environment silver nanoparticles will be released to the external media?
The current knowledge regarding silver nanoparticles/other polymers systems should be also given in the introduction. I do not understand why Carrageenan was chosen as a matrix for Ag-nanoparticles.
In my opinion simultaneously to SEM imaging also EDX analysis should be performed to evaluate the Ag distribution within the polymer matrix. Was there any difference between the Ag-nanoparticles content between both film surfaces? Is there any evidence of Ag sedimentation/separation in Ag/polymer system during the film formation process?
Please refer to the Mie theory and give the size of Ag nanoparticles.
Based o 2.1.4: PSA or SEM/TEM results are “more realistic”
The overall discussion of the mechanical properties is really laconic and should be improved.
“However, FT-IR spectrum of the nano-composite hydrogel film shows an absorption band in 400 cm−1 which can be related to the presence of silver nanoparticles (Figure 6B) [21, 34].” – I do not see such a band in FTIR spectra. The FTIR discussion should be improved
“The composite has thermal stability higher than the κ-carrageenan polysaccharide which might be related to the reaction and crosslinking of the κ-carrageenan and silver nanoparticles.” – explain how it is possible. Moreover, explain why the thermal behavior within 20-800 °C need to be tested? How it influences the possible application of this material? A discussion of TGA results should be improved.
Fig. 8A – are these really pores? (see Figure 8C) or these are small particles observed on the surface? Where the samples sputtered with any metal before imaging? As Fig. 8C was recorded for higher magnification pores (suggested to be present based on Figure 8a) should be also visible, but they are not.
How the discussion provided (lines 192-198) refers to the possible way of acting of silver nanoparticles in polymer matrix? Are these nanoparticles released from this matrix? If yes, are they still in the final, cleaned water? In my opinion, the statement, that such a system can be used at homes is highly overestimated.
Conclusions are rather summary of particular chapters – should be improved.
Particular comments:
Statement (lines 38-40): “The antimicrobial activity of 38 k-Carrageenan/silver nanoparticles against Gram positive, Gram negative and yeast pathogens was 39 100% effective.” is overestimated
Line 30: remove “primitive”
Line 45” “The recent technological fields” – too general statement
Line 52: “higher” (in comparison to?) or “high”
Line 53-54: describe both methods shortly, especially green methods, especially in the context of sentences provided in lines 55-57
Line 57: correct “[10] [11].”
Lines 71-72: sentence unclear
Line 87” analysis was performed not investigated
Provide Fig. 3 and 4 in higher resolution as I cannot see the scale (especially in Fig. 3c)
Compare size distribution given in 2.1.2 with the results calculated based on 2.1.1
Lines 103-105: change “amplification” into “magnification”
The X-axis in Figure 7 should end at 100%.
Line 172: Add number 2.2.5
Indicate what pathogenic are mostly found in water.
Line 204: add number 2.2.6
Line 214: add also other reagents used in this study
Line 222: change the title as it refers also to Ag-nanoparticles formation
Line 228: change “water dissolved contents” into water-soluble contents”
Please indicate the thickness of obtained polymeric films
Line 284: correct the sentence “The testing……to be tested…..”
Others:
“havemany” (line 66), “figure 3” (line 111), “figure 4” (line 108), check in all manuscript “k-carrageenan” vs. “κ-carrageenan”, “band in…” – should be “at”, remove brackets (line 150-151, Figure 7), “figure 8A” (line 161), “thee” (line 194), “roller membranes” – correct the formal name of such a membrane system,